# Efficient Quantification of Multimodal Interaction at Sample Level

Zequn Yang [1]   Hongfa Wang [2 3]   Di Hu [1 4 5]

## Abstract

Interactions between modalities—redundancy, uniqueness, and synergy—collectively determine the composition of multimodal information. Understanding these interactions is crucial for analyzing information dynamics in multimodal systems, yet their accurate sample-level quantification presents significant theoretical and computational challenges. To address this, we introduce the *Lightweight Sample-wise Multimodal Interaction* (LSMI) estimator, rigorously grounded in pointwise information theory. We first develop a redundancy estimation framework, employing an appropriate pointwise information measure to quantify this most decomposable and measurable interaction. Building upon this, we propose a general interaction estimation method that employs efficient entropy estimation, specifically tailored for sample-wise estimation in continuous distributions. Extensive experiments on synthetic and real-world datasets validate LSMI's precision and efficiency. Crucially, our sample-wise approach reveals fine-grained sample- and category-level dynamics within multimodal data, enabling practical applications such as redundancy-informed sample partitioning, targeted knowledge distillation, and interaction-aware model ensembling. The code is available at https://github.com/GeWu-Lab/LSMI_Estimator.

## 1. Introduction

Multimodal data offer richer information sources, significantly enhancing information acquisition capabilities. This richness largely derives from inter-modal interactions, the harnessing of which is crucial for advancing multimodal models. Traditionally, research has focused on capturing *redundant* interactions by aligning modalities and extracting consistent information among them (Rahate et al., 2022). Such methods are particularly effective when information is entirely shared across modalities, as in the case of images paired with text captions (Radford et al., 2021). However, modality inconsistency can also contain valuable information. For instance, in video understanding, certain phenomena (e.g., wind blowing) may provide limited or even misleading information in the visual modality but can be clearly discerned through the auditory modality. This highlights the importance of extracting modality-specific *unique* information. Moreover, when individual modalities alone are insufficient to convey information, their combination can induce additional insights. For example, sarcasm can be detected through a positive expression paired with a negative tone. In such cases, multimodal information emerges from the collaboration of modalities, reflecting *synergistic* interactions. In summary, multimodal interactions – encompassing redundancy, uniqueness, and synergy – play critical roles in producing multimodal information and offer credible perspectives for advancing multimodal learning.

To better characterize the dynamic interplay of these multimodal interactions, defining and quantifying interactions in a principled manner is of significant importance. Initially, *Partial Information Decomposition* (PID) (Williams & Beer, 2010) is proposed as a field dedicated to investigating the nature of interactions from an information-theoretic perspective. Research on PID has primarily focused on providing formal definitions of interactions (Lizier et al., 2018; Mages & Rohner, 2023) within discrete distributions. Additionally, several studies have extended the concept of interaction decomposition to continuous Gaussian (Venkatesh et al., 2024) and low-dimensional (Pakman et al., 2021) distributions. Building on these foundations, Liang et al. (Liang et al., 2023b) propose a distribution-optimization-based strategy for interaction estimation. Their approach utilizes neural estimation to achieve interaction quantification at the distribution level, which has been successfully applied for dataset evaluation and model selection (Liang et al., 2023a). These advancements underscore the promising potential of interaction quantification in advancing multimodal research.

---

[1]Gaoling School of Artificial Intelligence, Renmin University of China, Beijing, China [2]Tencent Data Platform [3]Tsinghua University, Beijing, China [4]Beijing Key Laboratory of Research on Large Models and Intelligent Governance, Beijing, China [5]Engineering Research Center of Next-Generation Intelligent Search and Recommendation, MOE, China. Correspondence to: Di Hu <dihu@ruc.edu.cn>.

*Proceedings of the 42$^{nd}$ International Conference on Machine Learning*, Vancouver, Canada. PMLR 267, 2025. Copyright 2025 by the author(s).

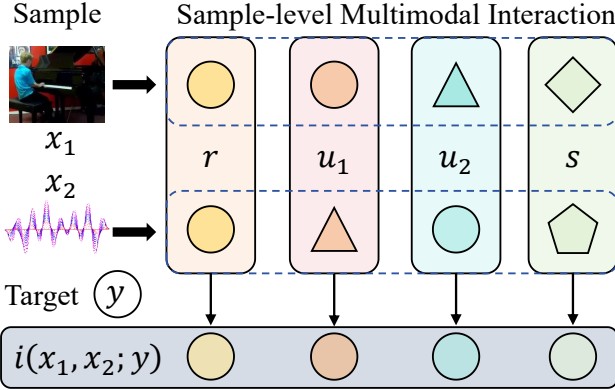

Figure 1: A brief illustration of multimodal interactions at the sample level, including **r**edundancy ($r$), **u**niqueness ($u_1, u_2$), and **s**ynergy ($s$), which collectively constitute multimodal information $i(x_1, x_2; y)$.

Beyond such quantifications, analyzing multimodal interactions at the individual sample level is both important and challenging. Given that interaction patterns can vary substantially across samples (Luppi et al., 2024), sample-level analysis provides a more granular understanding of these interaction patterns and enhanced interpretability for real-world applications (Lizier et al., 2013). However, existing distribution-level approaches (Bertschinger et al., 2014; Liang et al., 2023b) primarily quantify interaction information for an entire dataset, lacking the capability for such fine-grained, sample-level analysis. Although pointwise partial information decomposition approaches (Finn & Lizier, 2018a) have been proposed for sample-level interaction estimation, they struggle to provide efficient and practical solutions for continuous distributions. This limitation highlights the need for novel methods capable of realizing effective sample-level interaction quantification.

To address this challenge, we propose the *Lightweight Sample-wise Multimodal Interaction* (LSMI) estimation approach, which enables efficient and reliable quantification of sample-level interactions. As illustrated in Figure 2, our estimator aim to distinguish task-relevant information (circles) generated from two modalities, $x_1, x_2$, into redundant, unique, and synergistic interaction. We define interactions using pointwise information and further equip a lightweight model for efficient and precise measurement. On the one hand, we adopt a clear and credible definition of pointwise information decomposition using redundancy-based set-theoretic intuition. Given that pointwise mutual information can be negative, adhering to set-theoretic intuition remains a challenge. To overcome this, we partition mutual information into target-related information components, allowing for the measurement suitable for redundancy in each component while preserving the set-theoretic intuition of redundancy. On the other hand, we employ an efficient

sample-wise interaction estimation process based on this definition. By leveraging lightweight entropy estimation models (Pichler et al., 2022), we enable the efficient calculation of sample-wise redundancy interactions, as well as the identification of uniqueness and synergy interactions. Extensive experiments confirm the high efficiency, precision, and real-world applicability of our sample-level estimator. Our method also uncovers interaction dynamics at both sample and class levels. These insights not only enhance interpretability but also enable practical applications such as redundant data partitioning, interaction-guided knowledge distillation, and targeted model ensemble that effectively combine sub-models while preserving their specificity.

In summary, our contributions are as follows:

1. To the best of our knowledge, we are the first to explicitly quantifying multimodal interaction at the sample level for real-world data.

2. We propose a sample-level interaction estimation approach that ensures both precision and efficiency.

3. Our method reveals the underlying interaction dynamics within the data, facilitating a deeper understanding and providing guidance for multimodal learning.

## 2. Related Work

### 2.1. Multimodal Interaction Learning

Multimodal learning paradigms aim to extract various types of information embedded across different modalities. One key strategy is to learn consistency, thereby exploiting redundant (or shared) information. Consistency learning (Zhan et al., 2018; Rahate et al., 2022; Yang et al., 2021) and contrastive learning (Radford et al., 2021; Yuan et al., 2021) are common methods to capture this redundancy. Furthermore, gradient-based modulation techniques (Peng et al., 2022; Yang et al., 2025) can mitigate learning disparities across modalities (Yang et al., 2024), bolstering consistency indirectly and enhancing multimodal performance. Conversely, inconsistencies can highlight unique or synergistic information. Unique information arises when heterogeneous modalities inherently contain different levels or types of information (Gat et al., 2020; Zhang et al., 2023b); approaches here often enhance individual modality representations to isolate modality-specific contributions (Tsai et al., 2018; Wu & Goodman, 2018). Synergistic information emerges when the combination of modalities generates new insights not available from any single modality, especially when individual modalities are insufficient for the task. Capturing synergy typically involves more sophisticated fusion mechanisms and dynamic integration strategies (Fukui et al., 2016; Jayakumar et al., 2020). Understanding these fundamental interactions—redundancy, uniqueness, and syn-

ergy—is crucial for guiding the design and selection of multimodal learning methods. Therefore, this paper proposes an efficient analytical framework to characterize multimodal interactions, aiming to provide in-depth explanations and guidance for the field.

## 2.2. Interaction Quantification

Data from multiple sources often interact to reveal how information originates across these sources. Estimation of the interactions provides valuable insights into neural science (Wibral et al., 2017; Celotto et al., 2024). Partial Information Decomposition (Williams & Beer, 2010) serves as a foundational framework, describing three types of interaction: redundancy, uniqueness, and synergy. Subsequent work has provided different definitions of these interactions for discrete distributions (Knuth, 2019; Mages & Rohner, 2023), but these definitions are difficult to extend to continuous distributions in real-world scenarios. Some studies have extended these methods to continuous low-dimensional (Pakman et al., 2021) or Gaussian (Venkatesh et al., 2024) distributions. Additionally, Liang et al. (Liang et al., 2023b) have measured interactions in more complex real-world datasets by optimizing the distribution to determine uniqueness. However, measuring sample-wise multimodal interactions remains an open issue (Lizier et al., 2013; Liang et al., 2023b), and addressing this challenge could significantly enhance the fine-grained understanding of multimodal data and the learning preferences across modalities. To this end, this work proposes a practical approach for quantifying sample-wise interaction estimation.

## 3. Method

In this section, we introduce a lightweight sample-wise multimodal interaction estimation approach. We begin by providing a detailed explanation of interaction estimation, followed by a principled definition of sample-level interactions from the perspective of redundancy. Subsequently, we implement efficient interaction estimation at the sample level for continuous distributions.

### 3.1. Preliminary

In this study, we primarily analyze the mutual information between two modalities $X_1$ and $X_2$ with respect to a target variable $Y$, where samples $x_1, x_2$ and target $y$ can be considered as events from the joint distribution. Our framework can be naturally extended to handle multiple modalities (see Appendix subsubsection A.3.1 for details). For information metrics, we use the subscript $i$ to indicate pointwise mutual information, calculated as $i(x; y) = \log \frac{p(x,y)}{p(x)p(y)}$. The superscript $I$ denotes average mutual information, $I(X; Y) = \mathbb{E}_{x,y}[i(x; y)]$.

**Interaction Estimation** Interaction estimation, also referred to as Partial Information Decomposition, offers a comprehensive framework for understanding the information conveyed through multiple modalities. In the context of multimodal information shared by $X_1$ and $X_2$ with the target $Y$, Interaction Decomposition classifies this information based on its distribution across the different modalities. It distinguishes between the information that is shared across modalities (redundancy), that which is uniquely represented within each modality (uniqueness), and the information that emerges only when both modalities are present (synergy). As a result, it decomposes the total information into four distinct components: redundancy $R$ between $X_1$ and $X_2$, uniqueness $U_1$ in $X_1$, uniqueness $U_2$ in $X_2$, and synergy $S$, which emerges only when both $X_1$ and $X_2$ are jointly considered. It satisfies the following equation:

$$I(X_1; Y) = R + U_1, I(X_2; Y) = R + U_2,$$
$$I(X_1, X_2; Y) = R + U_1 + U_2 + S. \tag{1}$$

A widely recognized approach to interaction decomposition, based on the concept of uniqueness (Bertschinger et al., 2014; Liang et al., 2023b), determines interactions by identifying a *base distribution* that minimizes unique information. This method, inspired by decision-making theory, then leverages this base distribution to compute redundancy ($R$), uniqueness ($U$), and synergy ($S$). However, this distribution-level optimization presents two primary challenges. First, optimizing the distribution is computationally expensive. Second, its design is restricted to calculating interactions over the entire distribution, which impedes sample-wise estimation and, consequently, limits its interpretability and practical utility for fine-grained analyses.

### 3.2. Redundancy-based Interaction Framework

Compared with calculating interaction over the whole distribution, sample-level interaction estimation enables details information analysis towards each samples, which provide in-depth knowledge for multimodal interaction (Lizier et al., 2013). To achieve it, the multimodal information can be denoted in a pointwise way, in order to inspect how each event brings information about the target. Denote lower-script $r, u, s$ as pointwise interaction corresponding with $R, U, S$, we can extend Equation 1 to event-level:

$$i(x_1; y) = r + u_1, \ i(x_2; y) = r + u_2,$$
$$i(x_1, x_2; y) = r + u_1 + u_2 + s. \tag{2}$$

This equation involves four unknown variables: $r, u_1, u_2$, and $s$. Since the mutual information terms for unimodality $i(x_1; y), i(x_2; y)$ and multimodality $i(x_1, x_2; y)$ can be estimated using discriminators or neural estimation methods, the key challenge lies in determining the value of the remaining degree of freedom in Equation 2.

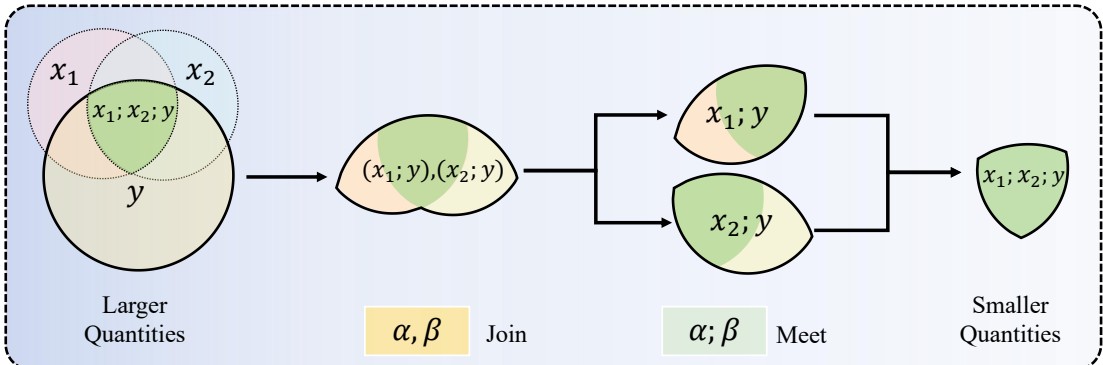

Figure 2: Event-level redundancy information estimation framework. The proposed measure ensures that information quantities monotonically decrease along the decomposition path (indicated by arrows), enabling precise quantification of redundant information components.

To address this challenge, we investigate redundancy from a pointwise perspective, aiming to provide a reliable and intuitive quantification measure. Redundancy, as a fundamental interaction type, represents the shared information across multiple data sources. By definition, it should not exceed the information present in any individual source. This property enables us to derive redundant interactions through a decomposition framework that eliminates non-redundant information components. Building on this principle, we develop an event-level decomposition framework for task-related information based on lattice structure. Let $\alpha$ and $\beta$ represent two distinct events, where $(\alpha, \beta)$ denotes their joint occurrence and $(\alpha; \beta)$ represents their shared component. This formulation facilitates precise measurement of inter-event relationships, forming the redundancy estimation framework depicted in Figure 2. The framework systematically traces information flow through each lattice point to identify redundant information components. An appropriate measure that satisfies this framework would then allow the information within the shared component, specifically $(x_1; x_2; y)$, to be quantified as redundancy.

With this framework, a significant challenge arises in applying a reasonable measure for the framework. The measure should satisfy the monotonic decrease of redundancy decomposition on each lattice (Figure 2 from left to right). An intuitive approach is to apply the pointwise information measure directly on this framework. However, as pointwise mutual information $i(x; y)$ can be negative (when $x$ provides misleading information about $y$), monotonicity is not always guaranteed. Concretely, according to the redundancy framework, the following inequality should hold:

$$i(x_1; y) \leq i((x_1; y), (x_2; y)) \leq i(x_1; y) + i(x_2; y). \quad (3)$$

However, the pointwise mutual information $i(x_2; y)$ can be negative, which would violate the inequality. Thus, the monotonicity over the redundancy framework cannot be consistently achieved directly under the information measure, which is highlighted by previous literature (Finn & Lizier, 2018b). Note that this conclusion differs from the results obtained when averaging over the entire distribution, as the average information satisfies $I(X_m; Y) \geq 0$ and can ensure monotonicity.

Hence, there is a need for an alternative feasible way for determining redundancy. Inspired by (Ince, 2017; Finn & Lizier, 2018a), we can apply redundancy on partial information component, and obtain the redundant information. In detail, we divide the information into two information components $i(x; y) = i^+(x; y) - i^-(x; y)$. This division should be uniquely determined given $x, y$, and both components should satisfy the monopoly condition and be positive. Accordingly, we use the following division:

$$
\begin{aligned}
i^+(x; y) &= h(x) = -\log p(x), \\
i^-(x; y) &= h(x|y) = -\log p(x|y).
\end{aligned}
\quad (4)
$$

Therefore, given the positive property, each component within this division adheres to monotonicity. As a result, both measures, $i^+$ and $i^-$, are compatible with the lattice on Figure 2. Consequently, we define the redundancies $r^+$ and $r^-$ on each component for

$$
\begin{aligned}
r^+(x_1; x_2; y) &= \min\left(i^+(x_1; y), i^+(x_2; y)\right) \\
r^-(x_1; x_2; y) &= \min\left(i^-(x_1; y), i^-(x_2; y)\right).
\end{aligned}
\quad (5)
$$

With this definition, we integrate redundancy from both components to obtain the event-level redundant interaction.

$$r(x_1; x_2; y) = r^+(x_1; x_2; y) - r^-(x_1; x_2; y), \quad (6)$$

which uniquely determines the multimodal interaction. This enables us to accurately determine the respective values of $u_1$, $u_2$, and $s$ using Equation 2. Furthermore, we can obtain the average interaction values $R$, $U_1$, $U_2$, and $S$ within the dataset by averaging over these sample-level interactions.

| Task | XOR | | | | OR | | | | XOR+NOT | | | |
|---|---|---|---|---|---|---|---|---|---|---|---|---|
| Interaction | $R$ | $U_1$ | $U_2$ | $S$ | $R$ | $U_1$ | $U_2$ | $S$ | $R$ | $U_1$ | $U_2$ | $S$ |
| PID-CVX | 0.000 | 0.000 | 0.000 | 0.692 | 0.210 | 0.001 | 0.000 | 0.342 | 0.000 | 0.000 | 0.338 | 0.346 |
| PID-Batch | 0.000 | 0.002 | 0.002 | 0.690 | 0.200 | 0.018 | 0.018 | 0.322 | 0.003 | 0.000 | 0.257 | 0.381 |
| LSMI (ours) | 0.000 | 0.001 | 0.001 | 0.691 | 0.215 | 0.001 | 0.000 | 0.345 | 0.000 | 0.000 | 0.336 | 0.347 |
| GT | 0.000 | 0.000 | 0.000 | 0.693 | 0.215 | 0.000 | 0.000 | 0.347 | 0.000 | 0.000 | 0.347 | 0.347 |

Table 1: Comparison of the proposed LSMI with previous interaction estimators on circuit logic (XOR, OR, and XOR+NOT).

---

**Algorithm 1** Lightweight Sample-wise Multimodal Interaction Estimation (LSMI) Algorithm

---

1: **Input:** Bimodal data $x_1, x_2$, target $y$; discriminative models $p(y|x_1, x_2), p(y|x_1), p(y|x_2)$.
2: **Initialize:** Entropy estimators $h_{\theta_1}(\cdot), h_{\theta_2}(\cdot)$.
3: Train entropy estimators $h_{\theta_1}, h_{\theta_2}$ using Equation 7 on data from $p(x_1), p(x_2)$ respectively.
4: Compute sample-wise $h(x_1), h(x_2)$ using $h_{\theta_1}, h_{\theta_2}$; then compute $h(x_1|y), h(x_2|y)$ via Equation 8.
5: Compute pointwise redundancy indicators $r^+, r^-$ via Equation 5; then redundancy $r \leftarrow r^+ - r^-$.
6: Compute pointwise $i(x_1; y), i(x_2; y), i(x_1, x_2; y)$ using $p(y|x_1), p(y|x_2), p(y|x_1, x_2)$; then derive interactions $u_1, u_2, s$ via Equation 2.
7: **Output:** Sample-wise interactions $r, u_1, u_2, s$.

---

### 3.3. Lightweight Interaction Estimation

Although we have defined the measure for redundancy decomposition well, how to quantify this interaction over continuous distributions and real-world datasets remains a question. In this work, we refer to the tool KNIFE (Pichler et al., 2022) as the differential entropy estimation, which is efficient and suitable for sample-wise estimation over complex distributions. Denote $h_\theta$ as entropy estimator,

$$\mathbb{E}[h_\theta(x)] = \mathbb{E}[h(x)] + D_{KL}(p(x)||p_\theta(x)) \geq H(X) \quad (7)$$

serves as an upper bound for entropy, as the Kullback-Leibler divergence ($D_{KL}$) is inherently positive. Therefore, the estimator can be optimized by tuning parameters $\theta$ to minimize the $D_{KL}$, thereby tightening this upper bound. Consequently, we derive $h_{\theta_1}(x_1)$ and $h_{\theta_2}(x_2)$ as estimations for the positive information component $i^+$, as specified in Equation 4. Additionally, once the unimodal discriminative models $p(y|x_1)$ and $p(y|x_2)$ are determined, the negative component $i^-$ can be estimated as follows:

$$i^-(x_m; y) = h_{\theta_m}(x_m) - h(y) - \log p(y|x_m), m \in [2]. \quad (8)$$

Details are shown in Algorithm 1. By adopting this method, we circumvent the necessity of modeling the base distribution to achieve interaction decomposition. Consequently, our approach demands only a limited number of parameters

for entropy estimation, thereby reducing time consumption typically associated with distribution modeling. Furthermore, our lightweight method facilitates the measurement of pointwise interactions, a capability not extendable to the sample level in previous approaches. Overall, this approach enhances flexibility and scalability, thereby providing insights and explanations for multimodal learning.

## 4. Experiment

### 4.1. Synthetic Experiment

#### 4.1.1. SETUP

In synthetic experiments, we validate the precision of our estimation approach using two typical distributions: a bitwise circuit system and a mixture of Gaussians, where interactions can be explicitly calculated. We also conduct experiments with manually controlled interactions to elucidate the effects of different interaction patterns. To better adapt to real-world data scenarios, additional noise is introduced into these datasets, making their data distributions continuous. The generation process of these datasets is illustrated in subsection A.1. Our estimation method is compared with the CVX (PID-CVX) estimator, designed for discrete distributions (Liang et al., 2023b), and the batch-level (PID-Batch) estimator, which is applicable to both discrete and continuous distributions (Liang et al., 2023b).

#### 4.1.2. CIRCUIT LOGIC

We first employ bitwise circuit logic systems validating precision. We construct samples using fundamental logic operations (e.g., OR, XOR, and a mixture of NOT and OR) combined with small Gaussian noise. The crucial advantage of this setup is its capacity to provide an objective and verifiable benchmark: the deterministic nature of logic operations allows for sound calculation of ground truth (GT) interactions (Bertschinger et al., 2014). Consequently, a closer alignment between predicted interactions and this GT directly signifies a more accurate estimation. We compare our method against PID-CVX and PID-Batch (Liang et al., 2023b). The experimental results in Table 1 demonstrate that our estimators accurately recover the correct interactions, closely matching the ground truth and showing favorable performance compared to these baseline methods.

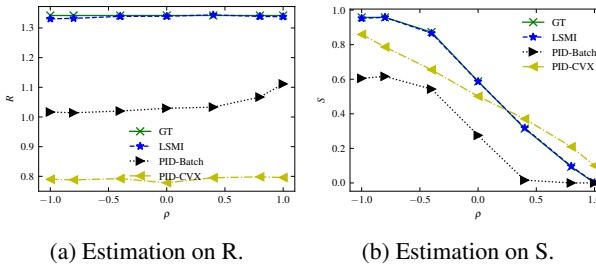

(a) Estimation on R.      (b) Estimation on S.

Figure 3: Comparison of estimators on data with a mixture of Gaussian distributions.

### 4.1.3. MIXTURE OF GAUSSIAN

For the simulation of a continuous distribution, we design a dataset based on a mixture of Gaussian distributions. Specifically, for both $x_1$ and $x_2$, the distributions are defined as:

$$p(x_m) = \sum_{y=1}^{K} \pi^{(y)} \mathcal{N}(x|\mu^{(y)}, \Sigma^{(y)}), \quad (9)$$

where different modalities share the same mean $\mu^{(y)}$ and covariance $\Sigma^{(y)}$. This distribution can be approximated as a $K$-way classification task, which is useful for measuring the interaction between $x_1, x_2$ and $y$.

To manipulate interactions between modalities, we impose constraints on the covariance of Gaussian distributions for the two modalities. Specifically, we adjust the covariance $\rho(x_1, x_2|y) = \{-1, -0.8, -0.4, 0, 0.4, 0.8, 1\}$ to obtain different joint distributions. The intended effect of these adjustments is that they only influence the amplitude of the synergy interaction, while the redundancy and uniqueness of interactions remain unchanged, as outlined by (Bertschinger et al., 2014). The experimental comparisons, detailed in Figure 3, begin with a ground truth (GT) calculated through a mixture of Gaussian model. Our findings indicate that both PID-CVX and PID-Batch fail to consistently estimate interactions, showing divergence from the ground truth despite exhibiting similar trends across different interaction types as $\rho$ changes. In contrast, our proposed method demonstrates a high capacity to closely match the ground truth interactions, showcasing the precision of our estimation approach.

### 4.1.4. PRESET INTERACTION

We also verify the precision of manually designed datasets with controllable interactions. In this dataset, we ensure that each sample contains one type of interaction—Redundancy, Uniqueness, or Synergy—and adjust the proportion of each sample to preset the interaction. Consequently, we mix these data in certain proportions (e.g., $\frac{1}{4}U + \frac{3}{4}S$ indicates that $\frac{1}{4}$ of the data follows Uniqueness while the remainder exhibits Synergy). The experimental results are presented in Figure 4. The ground truth of the radar is shown in Figure 4 (d), which is asymmetric due to our setting. Both PID-CVX

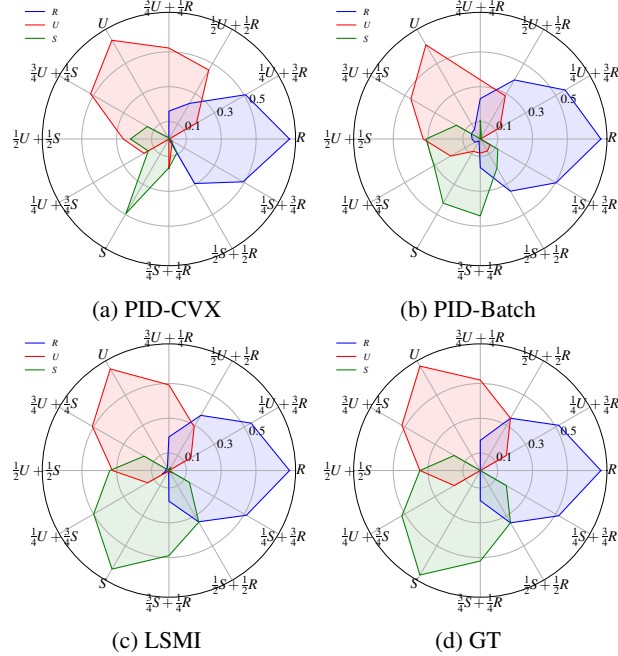

(a) PID-CVX      (b) PID-Batch

(c) LSMI      (d) GT

Figure 4: Comparison of estimators with preset interactions.

(Figure 4 (a)) and PID-Batch (Figure 4 (b)) show discrepancies in estimating Synergy interaction (within the green lines), whereas our method (Figure 4 (c)) handles these interactions effectively, enabling more precise estimations.

### 4.2. Validation on Real-world Datasets

We apply our LSMI approach to real-world datasets to demonstrate its capability of efficiently estimating multimodal interactions at the sample level and providing valuable insights into interaction modeling. Additional experiments, which include multiple modal analyses, architectural comparisons, and distribution shifting evaluations, are detailed in subsection A.3.

### 4.2.1. EXPERIMENTAL SETTING

**Dataset** We conduct experiments on extensive multimodal datasets encompassing various tasks and modalities. These include Food-101 (Bossard et al., 2014), which focuses on food classification using text and image modalities; CREMA-D (Cao et al., 2014), dedicated to emotion analysis with audio and visual modalities; Kinetic-Sounds (KS) (Arandjelovic & Zisserman, 2017), an action recognition task employing audio and visual modalities; UCF-101 (Soomro et al., 2012), a multimodal action recognition dataset utilizing RGB and optical flow modalities; CMU-MOSEI (Zadeh et al., 2018), which addresses binary sentiment analysis through video (including audio and visual) and text modalities; and UR-funny (Hasan et al., 2019), aimed at humor detection using video and text.

| Dataset | KS | | | | Food-101 | | | | UR-Funny | | | | CMU-MOSEI | | | |
| Interaction | $R$ | $U_1$ | $U_2$ | $S$ | $R$ | $U_1$ | $U_2$ | $S$ | $R$ | $U_1$ | $U_2$ | $S$ | $R$ | $U_1$ | $U_2$ | $S$ |
|---|---|---|---|---|---|---|---|---|---|---|---|---|---|---|---|---|
| PID-Batch | 3.16 | 0.02 | 0.19 | 0.01 | 4.23 | 0.24 | 0.00 | 0.14 | 0.02 | 0.03 | 0.01 | 0.06 | 0.18 | 0.34 | 0.02 | 0.03 |
| LSMI | 3.28 | 0.11 | 0.00 | 0.03 | 4.19 | 0.34 | 0.00 | 0.08 | 0.02 | 0.12 | 0.01 | 0.24 | 0.13 | 0.22 | 0.01 | 0.00 |
| Human | 2.32 | 1.61 | 1.45 | 0.48 | 4.06 | 0.92 | 0.05 | 0.00 | 2.30 | 2.73 | 2.33 | 2.50 | 3.27 | 3.37 | 2.87 | 1.03 |

Table 2: Comparison of average interaction over various real-world datasets.

| Interaction | Categories preferred by each interaction | | | | |
|---|---|---|---|---|---|
| $r$ | playing organ | playing bagpipes | playing keyboard | playing accordion | playing drums |
| $u_v$ | pushing lawnmower | shoveling snow | shuffling cards | dribbling ball | bowling |
| $u_a$ | blowing nose | laughing | tapping guitar | playing guitar | playing clarinet |
| s | ripping paper | tickling | tap dancing | blowing out | - |

Table 3: Demonstration of the categories that most prefer specific types of interactions on the KS dataset.

| Method | $R$ | $U_1$ | $U_2$ | $S$ |
|---|---|---|---|---|
| *Feature-level fusion* | | | | |
| Joint | 3.165 | 0.143 | 0.000 | 0.122 |
| MMIB | 3.284 | 0.113 | 0.000 | 0.030 |
| Bilevel | 2.604 | 0.552 | 0.000 | 0.277 |
| *Decision-level fusion* | | | | |
| Additive | 3.397 | 0.006 | 0.000 | 0.029 |
| Weighted | 3.399 | 0.010 | 0.000 | 0.024 |
| QMF | 3.400 | 0.002 | 0.000 | 0.032 |
| *Additional Regulation* | | | | |
| Mod-drop | 3.163 | 0.134 | 0.000 | 0.116 |
| Alignment | 3.372 | 0.015 | 0.000 | 0.040 |
| Recon | 2.984 | 0.311 | 0.000 | 0.139 |

Table 4: Comparison of interaction components across different multimodal learning methods on the KS dataset.

**Baseline** We adopt three primary types of multimodal learning paradigms, details are shown in subsection A.2.

*Feature-level fusion*: **Joint learning** (Baltrušaitis et al., 2018), **MMIB** (Mai et al., 2022), **Bilinear** (Fukui et al., 2016). *Decision-level fusion*: **Additive** ensemble (Liang et al., 2021), **Weighted** ensemble (Shao et al., 2024), **QMF**(Zhang et al., 2023a). *Additional Regulation*: **Mod-drop** (Hussen Abdelaziz et al., 2020), **Alignment** (Radford et al., 2021), **Rec** (Tsang et al., 2020).

### 4.2.2. DATASET INTERACTIONS

Our LSMI-estimate method can be employed to uncover the inherent multimodal interactions within datasets, which varies significantly across diverse tasks and data domains. Modeling the true data distribution of real-world datasets is inherently complex. Accurate multimodal interaction estimation requires models that faithfully capture the true data distribution. Hence, we identify and utilize trained models that approximate the underlying data distribution, as

evidenced by minimal generalization error. Unimodal and multimodal information are then extracted by probing these learned representations. The dataset-level interactions are subsequently determined by averaging the sample-level interactions obtained from these selected models. To validate the reasonability of our approach, we compare our method with continuous interaction estimation (PID-batch) and human judgment as references. Since there are no predefined interaction values for real-world continuous datasets, we follow previous literature that annotated several datasets, including CMU-MOSEI and UR-Funny, and use the same settings as those studies (Liang et al., 2023b) to construct human judgment for interactions on the KS and Food-101 datasets. The comparison results are shown in Table 2. We observe that our LSMI estimator is largely consistent with the PID-Batch method. Furthermore, in KS, Food-101, and MOSEI datasets, our method aligns with the top two highest interactions in terms of redundancy and uniqueness, respectively. Similarly, in the UR-Funny dataset, our method exhibits strong correlations with the expected interaction patterns. While *human-annotated scores (on a 0-5 scale) are not direct measures of information content*, we observe strong Pearson correlations between LSMI estimates and human judgments: 0.98 for redundancy and 0.95 for text uniqueness on Food-101 dataset, indicating significant alignment in interaction quantification.

### 4.2.3. CASE STUDY

Our proposed LSMI estimator enables us to observe interaction variations at the sample level, providing insights into interaction differences across groups or categories. Specifically, we compute the interaction preferences for each category by averaging the interactions of the samples within that category. We analyze the top categories with the highest estimated values for redundancy $r$, visual uniqueness $u_v$, auditory uniqueness $u_a$, and synergy $s$. The results are presented in Table 3. Notably, instruments, which are easily

| | UR-Funny | CMU-MOSEI | CREMA-D | KS | UCF-101 | Food-101 |
|---|---|---|---|---|---|---|
| Number of classes | 2 | 2 | 6 | 31 | 101 | 101 |
| LSMI (s) | 454.4 | 667.1 | 426.1 | 501.5 | 678.9 | 504.0 |
| PID-Batch(s) | 1700.5 | 3124.4 | 5876.5 | 21928.0 | 48576.6 | 59679.5 |

Table 5: Comparison of time cost (s) over real-world datasets.

distinguishable by both sound and image, exhibit a high level of redundancy. Categories associated with visual elements (e.g., grass, snow) tend to prefer the visual modality. In contrast, categories where auditory features are more distinctive (e.g., blowing nose) show greater uniqueness in the audio modality. For tasks with more complex recognition (e.g., tickling), single modalities may struggle to distinguish the categories, leading these tasks to rely more on synergy. This finding aligns with human cognition. Additionally, we conduct sample-wise case studies to compare human annotations with model-estimated interactions. The comparison, shown in subsubsection A.3.4, demonstrates that our interaction estimation aligns well with human recognition, further validating its accuracy.

### 4.2.4. INTERACTION MODELING COMPARISON

Our LSMI framework reveals distinct capabilities of various multimodal learning paradigms in modeling different information interaction types, as demonstrated in Table 4. Feature-level fusion methods exhibit comprehensive capabilities for learning diverse interactions, while decision-level fusion approaches specialize in capturing data redundancy. Notably, specific regulation techniques show targeted advantages of interaction modeling: alignment-based methods effectively enhance redundant interaction learning, and reconstruction-focused approaches improve modeling of modality-specific unique information.

### 4.2.5. TIME EFFICIENCY

We compare the time cost of interaction estimation between our method and the PID-Batch method. The key difference lies in the way the two methods handle interaction measurement. Our method is designed to measure entropy by learning the mapping from samples within each individual modality to pointwise entropy, i.e., $\mathcal{X}_m \rightarrow \mathbb{R}^n, m \in [2]$. In contrast, PID-Batch aims to model the entire distribution $\mathcal{X}_1 \times \mathcal{X}_2 \times \mathcal{Y} \rightarrow \mathbb{R}^n$, and then calculate the interaction based on this joint distribution. As shown in Table 5, our lightweight estimator demonstrates significantly higher efficiency in interaction estimation compared to PID-Batch. Specifically, the computational cost of PID-Batch scales with the number of classes, resulting in substantial time complexity when the category space is large (e.g., UCF-101 and Food-101). In contrast, our LSMI estimator avoids joint distribution modeling, ensuring consistent time efficiency across tasks, regardless of the number of classes.

| Data | KS | | | CREMA-D | | |
|---|---|---|---|---|---|---|
| | V+A | V | A | V+A | V | A |
| All | 0.854 | 0.818 | 0.727 | 0.795 | 0.684 | 0.725 |
| Low | 0.850 | 0.805 | 0.729 | 0.782 | 0.702 | 0.715 |
| High | 0.877 | 0.824 | 0.726 | 0.801 | 0.688 | 0.728 |

Table 6: Performance comparison of ImageBind model fine-tuned on complete dataset (All), low-redundancy subset (Low), and high-redundancy subset (High) across unimodal and multimodal settings.

### 4.3. Application

As our LSMI estimator has demonstrated both accuracy and efficiency, how such interaction estimation can offer novel perspectives and tangible benefits remains critical. In this section, we demonstrate how sample-level multimodal interaction estimation can be applied to downstream tasks to boost multimodal performance. Specifically, we explore its utility in two main directions: (1) On the learning side, we investigate how interactions can improve dataset partitioning and enable targeted model distillation; (2) On the inference side, we explore the design of efficient ensemble strategies leveraging interaction patterns.

### 4.3.1. TARGETED DATA PARTITION

Effective multimodal learning depend on the nature of training data, with data interactions playing a crucial role. Our LSMI method addresses this by explicitly measuring these interactions and providing a quantifiable metric. This metric enables data partitioning based on interaction patterns suitable for specific frameworks, allowing models to learn from more tailored data subsets and ultimately boosting their performance. In this work, we employ the ImageBind model (Girdhar et al., 2023), which aligns different modalities into a common space. Since this alignment process naturally maximizes shared information—reflecting redundant interactions—we categorize samples into high-redundancy (High) and low-redundancy (Low) subsets based on their modal redundancy. We fine-tune ImageBind on these distinct subsets using its contrastive loss. To validate learning quality, we use simple linear layers on the extracted ImageBind features for task performance evaluation. The results in Table 6 show performance across multimodal and unimodal settings. Our experiments demonstrate that fine-tuning on the high-redundancy subset enhances multimodal feature quality through improved modality alignment

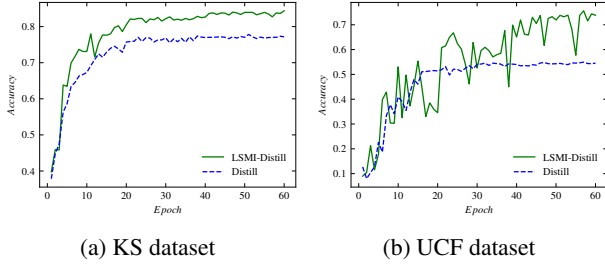

(a) KS dataset        (b) UCF dataset

Figure 5: Validation on LSMI-based distillation approach.

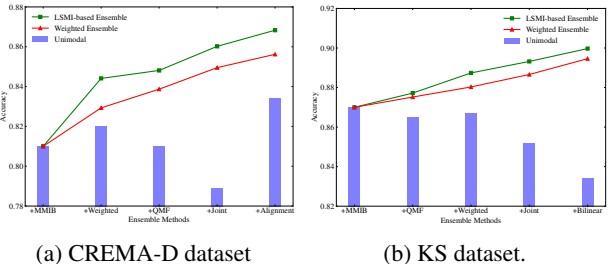

(a) CREMA-D dataset    (b) KS dataset.

Figure 6: Comparison between LSMI ensemble and weighted ensemble with various datasets.

facilitated by redundant interactions. This alignment primarily benefits information-rich modalities (e.g., Vision in KS, Audio in CREMA-D), enabling more effective learning while preventing misalignment with information-poor modalities. Conversely, the low-redundancy subset allows more thorough exploration of information-poor modalities (e.g., Audio in KS, Vision in CREMA-D), leading to their relatively stronger performance. These findings highlight the distinct roles of different data interactions in training and demonstrate the practical value of our sample-level interaction estimation for data partitioning.

### 4.3.2. INTERACTION-GUIDED DISTILLATION

Beyond its utility in interaction-based data partitioning, our method also offers valuable insights into model learning dynamics. LSMI can reflect how a multimodal model captures dynamic interaction patterns unique to each sample, enabling targeted interaction adjustment. To leverage these insights, we introduce an efficient knowledge distillation framework that utilizes sample-level interaction patterns - estimated by a high-performing teacher model - as informative signals to determine which knowledge components should be distilled into a student model trained from scratch. In particular: For redundancy (r) and uniqueness (u), we distill informative unimodal features. For synergy (s), we employ output-level distillation to capture cross-modal complementary effects. The distillation process is weighted by the relative magnitudes of r, u, and s. As a baseline comparison, we also evaluate a method that directly distills features from each individual modality, as illustrated in Figure 5. Our experimental results demonstrate that our method learns more effectively and acquires a greater amount of

informative knowledge compared to direct distillation. This indicates that our targeted distillation approach facilitates the learning of specific and distinctive information, enhancing the model's overall performance.

### 4.3.3. INTERACTION-GUIDED MODEL ENSEMBLE

When several well-trained models are available, an important question arises: how can we leverage interaction metrics to exploit the strengths of different models? Sample-level interactions can serve as indicators to measure the differences in how models extract information from individual samples, shedding light on a more reliable ensemble approach. In this study, we conduct experiments with several well-trained, yet diverse, multimodal approaches. We compute the interaction for each sample by averaging the interactions across different approaches and then ensemble each unimodal model with the corresponding interaction, which we call the LSMI-based ensemble. For comparison, we also implement a weighted ensemble and provide details on the unimodal accuracy. As shown in Figure 6, we demonstrate that even when the added models have lower accuracy than the original model, they still contribute significantly to performance improvements. This is because different models tend to focus on different interaction patterns, and ensembling based on interaction results in capturing more accurate interactions. This explains why LSMI-based ensemble methods outperform simple weighted ensembles.

## 5. Conclusion

This work aims to estimate the multimodal interaction of each sample, clarifying the quantities of interaction in redundancy, uniqueness, and synergy. We propose a lightweight entropy-based multimodal interaction estimation approach for efficient and precise sample-wise interaction measurement across various continuous distributions. We demonstrate the precision and efficiency of our estimation, as well as the utility of this sample-level estimation for guiding and improving multimodal learning. Our estimation more accurately reveals the information generation within the data, offering finer-grained insights that empower data-driven dataset construction and sample-specific algorithm design.

**Future work** This work on estimating sample-level interactions opens several avenues for future research, including: (1) investigating how models dynamically capture interactions within complex fusion mechanisms to elucidate adaptive learning and synergistic information generation; (2) uncovering relationships between training strategies and learned modal interactions to optimize multimodal learning systems; and (3) exploring how interaction estimation can enhance multimodal representation learning and modality-specific information acquisition, thereby enabling more fine-grained and dynamic interactive learning.

## Acknowledgement

This work is sponsored by CCF-Tencent Rhino-Bird Open Research Fund, the National Natural Science Foundation of China (Grant No.62106272), the Public Computing Cloud of Renmin University of China, and the fund for building world-class universities (disciplines) of Renmin University of China.

## Impact Statement

This paper presents work aimed at advancing the field of multimodal learning. There are many potential societal consequences of our work, none of which we feel must be specifically highlighted here.

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

# A. Experiments

## A.1. Synthetic Data Generation

In subsection 4.1, we employ three types of synthetic datasets to support our study. The first type is a circuit logic dataset, described in subsubsection 4.1.2, which is based on Boolean logic operators. The second type is a mixture of Gaussians with varying correlation coefficients ($\rho$), designed to simulate scenarios where unimodal distributions remain constant while multimodal distributions change, as detailed in subsubsection 4.1.3. The third type is a preset interaction dataset, where each dataset is constructed to include one or two predefined types of interactions, enabling the study of specific interaction dynamics, as outlined in subsubsection 4.1.4. Since the interactions in logic data can be easily determined (Bertschinger et al., 2014), this dataset provides a reliable ground truth for precision validation.

The circuit logic dataset, the first type in our study, is based on three fundamental Boolean logic operations: *OR*, *XOR*, and *NOT*. Each sample contains only one type of logic operation. For cases involving multiple logic operations (e.g., *XOR+NOT*), each logic type is equally represented in the dataset. To increase complexity, we add noise to each input dimension, requiring the evaluation model to denoise the input variables before estimating the corresponding logical relationships. As interactions in logic data can be precisely determined (Bertschinger et al., 2014), this dataset provides a reliable ground truth for validating our approach's precision.

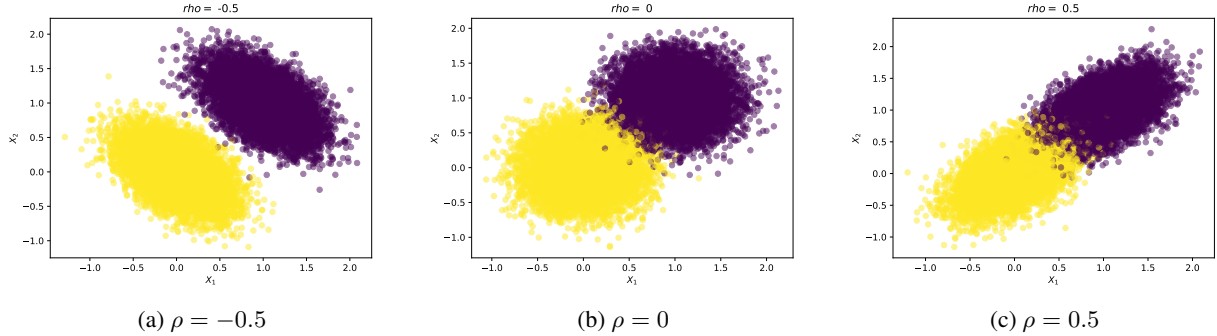

(a) $\rho = -0.5$          (b) $\rho = 0$          (c) $\rho = 0.5$

Figure 7: Illustration of bivariate Gaussian distributions with different correlation coefficients ($\rho$). From left to right: negative correlation ($\rho = -0.5$), no correlation ($\rho = 0$), and positive correlation ($\rho = 0.5$). These visualizations demonstrate how varying correlation affects the joint distribution while keeping marginal distributions constant.

The second type of dataset, a mixture of Gaussians, is designed to reflect variations in interactions when unimodal distributions remain constant while multimodal distributions change. Specifically, each individual modality follows a mixture of Gaussian distributions:

$$p(x_m) = \sum_{y=1}^{K} \pi^{(y)} \mathcal{N}(x | \mu^{(y)}, \Sigma^{(y)}), \tag{10}$$

where the unimodal distributions are identical across modalities. To simulate different interaction patterns, we adjust the joint distribution by setting the covariance $\rho(x_1, x_2 | y)$ to $\{-1, -0.8, -0.4, 0, 0.4, 0.8, 1\}$. For illustration, we use a two-component Gaussian mixture model to compute the means and variances (see Figure 7). We then apply PID decomposition (Bertschinger et al., 2014) to estimate the ground truth. When the unimodal distributions are held constant, the redundant ($R$) and unique ($U$) components remain unchanged, while only the synergistic ($S$) component varies, as demonstrated in the results of Figure 3.

For the third type, the dataset contains preset interactions, allowing for selective design of interactions to evaluate the capabilities of different methods under varying interaction patterns. In this synthetic data, each sample exhibits only a specific type of interaction, requiring the interaction estimation model to model the underlying distribution of each sample. As illustrated in Figure 4, each dataset is composed of samples exhibiting one or two types of interactions. The proportion of each interaction type is quantified using fractional notation, such as $\frac{1}{4}U + \frac{3}{4}R$. This indicates that $\frac{1}{4}$ of the samples display **Unique** interactions, while the remaining samples demonstrate **Redundant** interactions. The data generation process is executed in two sequential steps. First, the type of interaction for each sample is determined. Next, different interactions are mapped into high-dimensional data through linear transformations. For redundancy, both modalities are

| UCF-101 Dataset | | | | | CMU-MOSEI Dataset | | | | |
|---|---|---|---|---|---|---|---|---|---|
| Modality Pair | $R$ | $U_1$ | $U_2$ | $S$ | Modality Pair | $R$ | $U_1$ | $U_2$ | $S$ |
| Visual–OF | 2.111 | 2.483 | 0.000 | 0.000 | Vision–Text (V–T) | 0.121 | 0.000 | 0.163 | 0.005 |
| Visual–Diff | 3.474 | 1.121 | 0.000 | 0.000 | Vision–Audio (V–A) | 0.116 | 0.010 | 0.000 | 0.012 |
| OF–Diff | 1.998 | 0.003 | 1.476 | 0.239 | Audio–Text (A–T) | 0.127 | 0.000 | 0.248 | 0.002 |

Table 7: Pairwise interaction analysis on the UCF-101 and CMU-MOSEI datasets, with different modality combinations.

mapped simultaneously. For uniqueness, the non-informative modality is replaced with Gaussian noise. For synergy, an XOR-like construction is employed, ensuring that information is only effective when both modalities are present. To align with real-world scenarios, additional Gaussian noise is introduced into the high-dimensional data. This setup facilitates the experimental comparisons presented in Figure 4.

## A.2. Baseline

We adopt three primary types of multimodal learning paradigms:

**Feature-level fusion**: Integration of multiple modalities at the feature level. This includes: **Joint learning** (Baltrušaitis et al., 2018): A traditional paradigm where features from different modalities are concatenated and jointly mapped into the target space. **MMIB (Multimodal Information Bottleneck)** (Mai et al., 2022): Application of the variational information bottleneck principle to feature fusion. **Bilinear** (Fukui et al., 2016): Incorporation of dynamic interactions with learnable weights.

**Decision-level fusion**: Integration of unimodal predictions from different modalities. This approach includes: **Additive** (Liang et al., 2021): Ensemble of predictions by averaging. **Weighted** (Shao et al., 2024): Ensemble of predictions with pre-learned weights. **QMF**(Zhang et al., 2023a): Dynamic learning for each unimodality and learning weights.

**Additional Regulation**: Implementation of supplementary regulations to enhance multimodal learning: **Moddrop** (Hussen Abdelaziz et al., 2020): Application of dropout to partial modalities to prevent overfitting. **Alignment** (Radford et al., 2021): Introduction of contrastive loss to align modalities. **Rec** (Tsang et al., 2020): Application of unimodal reconstruction loss to strengthen unimodal capabilities.

## A.3. Additional Experiments

### A.3.1. EXTENSION TO MULTIPLE MODALITIES

Analyzing interactions involving more than two modalities presents considerable challenges in quantifying their complex relationships, such as the mutual information shared by three or more sources concerning a target task. A significant barrier is that established theoretical PID frameworks, designed for decomposing two-way interactions into synergistic, redundant, and unique components, do not directly or uniquely extend to these higher-order scenarios (Mages & Rohner, 2023; Williams & Beer, 2010). This is primarily because the information among more than three variables (including the various modalities and the target) become substantially more intricate.

Given the theoretical limitations in decomposing interactions beyond two modalities (Mages & Rohner, 2023; Williams & Beer, 2010), we adopt the pairwise interaction analysis strategy, as detailed in Appendix C.4 of (Liang et al., 2023b). This method systematically examines interactions between each pair of modalities. We applied this approach to the UCF-101 dataset (comprising Vision, frame difference (Diff), and optical flow (OF)) and the CMU-MOSEI dataset (Vision, Audio, and Text). The quantitative results of these pairwise interactions are presented in Table 7.

For the UCF-101 dataset, the analysis in Table 7 reveals several key patterns. Vision emerges as the most informative modality in terms of unique contributions ($U_1 = 2.483$ when paired with OF, and $U_1 = 1.121$ when paired with Diff). A strong redundant relationship is observed between Vision and Diff ($R = 3.474$). In contrast, the OF–Diff pair exhibits notable synergy ($S = 0.239$). This synergy is likely due to OF and Diff individually conveying less task-relevant unique information ($U_{\text{OF}} = 0.003$ and $U_{\text{Diff}} = 1.476$ in their pair, as $U_1$ and $U_2$ respectively), thus benefiting from combining their complementary aspects.

On the CMU-MOSEI dataset, Table 7 indicates that Text is the primary modality, consistently showing the highest unique

| Sample | Clean Label | | | | | Noisy Label | | | | |
|---|---|---|---|---|---|---|---|---|---|---|
| | $r$ | $u_1$ | $u_2$ | $s$ | Total | $\hat{r}$ | $\hat{u_1}$ | $\hat{u_2}$ | $\hat{s}$ | Total |
| #1 | 4.615 | -0.005 | 0.000 | 0.005 | 4.615 | -13.719 | 5.495 | 0.000 | -5.582 | -13.806 |
| #2 | 0.048 | 4.567 | 0.000 | -0.001 | 4.614 | -1.856 | -11.036 | 0.000 | -0.581 | -13.473 |
| #3 | 2.466 | 0.000 | 2.148 | 0.001 | 4.615 | -13.806 | 2.933 | 0.000 | -2.933 | -13.806 |

Table 8: Comparison of LSMI interaction estimation between samples with clean and noisy label.

| Dataset | Distribution | $R$ | $U_1$ | $U_2$ | $S$ | Total |
|---|---|---|---|---|---|---|
| UCF | ID | 3.319 | 1.289 | 0.000 | 0.006 | 4.614 |
| | OOD | 2.511 | 0.504 | 0.053 | 0.698 | 3.766 |
| KS | ID | 2.371 | 0.031 | 0.730 | 0.300 | 3.432 |
| | OOD | 1.864 | 0.083 | 0.386 | 0.559 | 2.892 |

Table 9: Comparison of LSMI-estimated multimodal interactions between In-Domain (ID) and Out-of-Distribution (OOD) data across different datasets.

information ($U_2 = 0.163$ in the Vision–Text pair, and $U_2 = 0.248$ in the Audio–Text pair). Vision and Audio generally exhibit lower unique contributions (e.g., $U_{\text{Vision (V–T)}} = 0$, $U_{\text{Audio (A–T)}} = 0$). However, we observe synergistic effects in certain modality pairs, most notably between Vision and Audio (V–A, $S = 0.012$), indicating that despite their relatively weak individual contributions, these modalities can provide complementary information when effectively combined.

### A.3.2. INTERACTION ESTIMATION ON DOMAIN SHIFTING

Previous experiments primarily involved datasets under stable domain conditions. A critical aspect, however, is to understand how these interactions behave under domain shifts. We investigate this under two primary settings: (1) label noise, where the ground-truth labels of multimodal samples are deliberately corrupted, and (2) a comparison of interactions in In-Domain (ID) versus Out-of-Distribution (OOD) scenarios.

Introducing label noise, which directly modifies the task definition, significantly alters the measured multimodal information dynamics, as detailed in Table 8. This alteration can even cause an inversion of the informational roles of different interaction components. For instance, an interaction component (e.g., redundancy or unique information from a modality) that was highly informative under clean labels might yield substantial negative information (i.e., become misleading) when labels are noisy. This occurs because the learned associations from clean data become counterproductive for the task defined by noisy labels, where original samples may now provide detrimental evidence.

When comparing In-Domain (ID) and Out-of-Distribution (OOD) scenarios (Table 9), OOD samples inherently provide less directly usable information. This is because models are typically not adequately trained on such data, leading to a diminished capacity to establish a robust mapping between the OOD inputs and the target task. Notably, we observe that synergistic information ($S$) tends to be more crucial for OOD performance. This suggests that when confronted with unfamiliar data, particularly when individual modalities alone are insufficient for the task, the model increasingly relies on the complementary strengths derived from combining different modalities. This underscores the heightened importance of synergy for better generalization, as the model must effectively integrate potentially weaker or less familiar signals from individual modalities to achieve the desired outcome.

### A.3.3. IMPACT OF FUSION STAGE ON LEARNED INTERACTIONS

To investigate how the fusion strategy affects learned multimodal interactions, we applied LSMI to a Hierarchical Multimodal Transformer architecture (Xu et al., 2023) on KS dataset. The specific model variant employed in our experiments features unimodal branches, each consisting of four Transformer layers. In such architectures, modalities are typically processed through these separate unimodal pathways, with each modality undergoing $l$ layers of dedicated processing before their representations are fused. In our experiments, we systematically varied the fusion point by adjusting $l$, the number of these unimodal layers preceding the first cross-modal interaction. A smaller $l$ (e.g., $l = 0$, indicating fusion at the input level)

| Unimodal Layers | $R$ | $U_1$ | $U_2$ | $S$ | Total |
|---|---|---|---|---|---|
| 0 | 1.238 | 0.737 | 0.000 | 1.445 | 3.420 |
| 1 | 1.844 | 1.011 | 0.000 | 0.566 | 3.421 |
| 2 | 1.975 | 1.093 | 0.000 | 0.355 | 3.423 |
| 3 | 2.299 | 0.870 | 0.000 | 0.250 | 3.419 |
| 4 | 2.335 | 0.907 | 0.000 | 0.181 | 3.423 |

Table 10: LSMI-based interaction analysis in a Hierarchical Multimodal Transformer (Xu et al., 2023), varying the fusion stage. $l$ denotes the number of unimodal layers processed before cross-modal fusion is introduced. $l = 0$ represents the earliest fusion (at input), and $l = 4$ represents the latest fusion (after 4 unimodal layers).

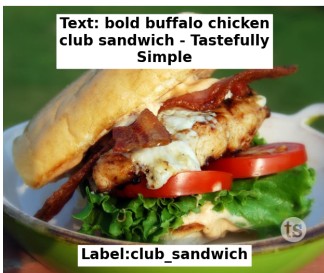 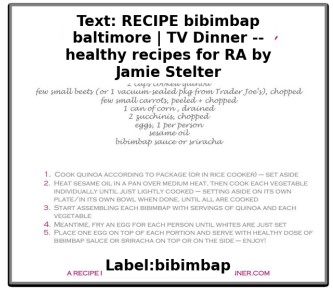 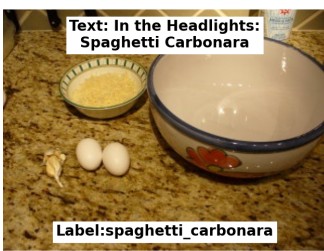

(a) $r = 4.6, u_t = 0, u_i = 0, s = 0$     (b) $r = -1.5, u_t = 6.1, u_i = 0, s = 0$     (c) $r = 4.2, u_t = 0.4, u_i = 0, s = 0$

Figure 8: Case studies on the Food-101 dataset showing different interaction patterns between visual and textual modalities.

corresponds to earlier fusion, while a larger $l$ (e.g., $l = 4$, signifying fusion after all 4 unimodal layers) represents later fusion.

The results, presented in Table 10, demonstrate a clear trend in the nature of learned interactions. While the total information captured remains relatively consistent across different fusion stages, the composition of interaction patterns varies significantly. Early fusion strategies (i.e., smaller $l$) tend to foster greater synergy. For instance, when fusion occurs at the input ($l = 0$), the learned synergy ($S = 1.445$) is more prominent than redundancy ($R = 1.238$). Conversely, as the fusion point is delayed to later stages (i.e., larger $l$), the model increasingly learns redundant information. With fusion after 4 unimodal layers ($l = 4$), redundancy ($R = 2.335$) significantly outweighs synergy ($S = 0.181$). These findings suggest that introducing cross-modal connections early in the architecture encourages the model to identify and combine novel, complementary information from different modalities, thereby capturing synergy. In contrast, later fusion appears to focus more on integrating information that has already been extensively processed within each modality, leading to a higher focus on redundant and overlapping information.

### A.3.4. CASE STUDY

To validate the effectiveness of our pointwise method, we evaluate interaction metrics at the sample level by selecting several representative samples and models. We conduct experiments on the Food-101 dataset and present a few illustrative cases in Figure 8. As shown in Figure 8, distinct interaction patterns are observed: In Figure 8 (a), a strong redundancy is evident, where the image and text consistently describe a sandwich. In Figure 8 (b), the image information is missing, and only the text provides meaningful information. In Figure 8 (c), partial redundancy is observed, with the text containing richer information than the image. These observations align with our interaction assessments, demonstrating the capability of our method to capture nuanced multimodal interactions.

