# OpenReview forum: "Efficient Quantification of Multimodal Interaction at Sample Level"
_ICML.cc/2025/Conference — ICML 2025 poster_

### Official Review · Reviewer_r5Mr · 2025-03-01

**Overall Recommendation:** 4

**Summary:**

In this paper, the authors presented a novel method for efficient quantification of multimodal interaction for a single multimodal sample. Pointwise mutual information may be negative, so monotonicity over the existing redundancy framework does not hold for single samples; therefore, the authors proposed an alternative method for determining redundancy: dividing the information into two components, where each component adheres to monotaniticity, and redundancy is calculated on each component before summed together. The authors further proposed a lightweight method to compute each component for a single sample. The authors evaluated their new algorithm on both synthetic dataset and real datasets, demonstrating that their method accurately estimates redundancy, uniqueness and synergy single samples. The authors further showed that their method is time-efficient, and can be used to improve multimodal ensemble and distillation.

**Claims And Evidence:**

The claims in the submission are supported clearly with empirical evidence.

**Essential References Not Discussed:**

N/A

**Experimental Designs Or Analyses:**

I have verified the soundness/validity of the experiments, including the evaluation over synthetic datasets, real-world datasets, time efficiency experiments, and application experiments (showing that LSMI results in better distillation and ensemble performance). The experiment design are sound.

**Methods And Evaluation Criteria:**

The proposed methods makes sense.

**Other Comments Or Suggestions:**

Minor problem: it seems like the citation for Liang 2023a and Liang 2023b are referring to the same paper in the bibliography.

**Other Strengths And Weaknesses:**

The paper is well written and well presented. The explanation of the key idea and methodology is clear.

**Questions For Authors:**

N/A

**Relation To Broader Scientific Literature:**

Previous works in multimodal machine learning demonstrated the importance of quantifying and extracting unique information from each modality as well as cross-modal interactions. While previous works have successfully quantified multimodal interactions over a data distribution and applied them to improve model performance, this work is the first to propose a method that can quantify multimodal interaction on a single data sample. Being able to quantify single-sample multimodal interaction would significantly improve fine-grained understanding of multimodal data, and this paper has also demonstrated that the proposed method can be used to improve multimodal distillation and ensemble.

**Theoretical Claims:**

Checked the correctness of the formulation of the new method (estimating the redundancy by dividing the information into two parts). Seems correct to me.

---

> ### Author Rebuttal · Authors · 2025-04-01
>
> We sincerely thank the reviewer for the positive feedback and constructive comments on our work. We have carefully addressed the citation issue mentioned in the review, and examine other reference and formualtions carefully. Thank you again for your time and effort.

---

> > ### Comment · Reviewer_r5Mr · 2025-04-01
> >
> > Thanks for your response. My review remains positive.

---

> > > ### Author Response · Authors · 2025-04-02
> > >
> > > Thank you for your positive feedback. We will carefully incorporate your valuable suggestions to improve our manuscript.

---

### Official Review · Reviewer_sN2h · 2025-03-10

**Overall Recommendation:** 4

**Summary:**

The paper proposes a novel framework to resolve conflicts between pointwise mutual information (PMI) and Partial Information Decomposition (PID) axioms in multi-modal learning. Traditional PID axioms (non-negativity, monotonicity) are defined for distribution-level redundancy but conflict with sample-level PMI, which can be negative. The authors address this by decomposing PMI into two non-negative components.

**Claims And Evidence:**

The paper’s ​theoretical claims (axiom compliance, decomposition structure) are well-supported by proofs. And the experiments

**Essential References Not Discussed:**

No

**Experimental Designs Or Analyses:**

Yes. For example, the experiments on distillation and dynamic fusion task demonstrate the effect of LSMI.

**Methods And Evaluation Criteria:**

Yes. Its ​practical and interpretability claims are well-supported by the experiments on synthetic data, distillation experiments, and dynamic fusion experiments.

**Other Comments Or Suggestions:**

see the weaness

**Other Strengths And Weaknesses:**

Strength:

​Innovative PMI decomposition strategy: Decomposing PMI into r+ and r− to resolve the PID axiom conflict is novel in the field of information theory and PID. Existing work usually avoids the problem of sample-level negative PMI, while this paper directly reconciles the contradiction between theory and practice through decomposition.

​Separation of sample and distribution-level redundancy: For the first time, the mathematical treatment of sample-level and distribution-level redundancy is clearly distinguished, which expands the scope of the application of PID.

Weaknesses:

​1. Visualization and examples: There is a lack of diagrams (such as the distribution of r+/r− in synthetic data) or examples (such as the redundant decomposition results of specific samples) to assist understanding.

2. How is the method scalable to three or more modes? Is there any experimental analysis?

3. i(x; y) = i+(x; y) − i−(x; y) in the derivation process may also be less than 0. So will there be samples with i(x1;x2; y)<0 in the actual application process? What is the physical meaning behind it?

**Questions For Authors:**

see the weaness

**Relation To Broader Scientific Literature:**

Existing methods focus on the distribution-level interaction quantification, this work extends them to the sample-level interaction quantification.

**Theoretical Claims:**

Yes.

---

> ### Author Rebuttal · Authors · 2025-04-01
>
> Thank you for their valuable suggestions. Here are our responses to the questions.
>
> **Q1: Visualization of $r^+$ and $r^-$**
>
> A1: Thank you for this valuable suggestion. To demonstrate the function of the $r^+$ and $r^-$ components derived from the total redundancy $r$, we conducted experiments on a Gaussian Mixture Model (GMM) dataset. We visualize the distributions of $r$, $r^+$, and $r^-$ in the provided [figure a](https://anonymous.4open.science/r/Submission2900-7FB8/Fig_a.png).
>
> As indicated by the distributions, $r^+$ appears to capture the core information consistently present across modalities within a sample, reflecting the minimum entropy along the unimodal marginal distributions (i.e., the most reliably available information from at least one modality). It represents the inherent information content shared by the modalities. In contrast, $r^-$ quantifies the portion of redundancy where individual modalities might struggle or provide conflicting cues regarding the task. It reflects the ambiguity or potential for error introduced if relying on the worse modality.
>
> **Q2: Scalability to three modality**
>
> A2: Thank you for this thoughtful suggestion. Extending interaction analysis on more than two modalities inherently involves quantifying complex interactions, such as the mutual information shared among three modalities and the target task. However, a key challenge arises because the clear theoretical definitions used to decompose two-way interactions into distinct synergistic, redundant, and unique contributions do not readily or uniquely translate to these higher-order cases [1,2].
>
> Given this theoretical limitation, we adopt the widely-used pairwise analysis approach (Appendix C.4 of [3]). This method systematically examines interactions between each pair of modalities. We implemented this approach on the UCF-101 dataset, analyzing the interactions among its three modalities: Vision, frame difference (Diff), and optical flow (OF). The quantitative results of these pairwise interactions are presented in the following Table.
>
> | modalityPair | $R$ | $U_1$ | $U_2$ | $S$ |
> |---|:---:|:---:|:---:|:---:|
> | Visual_OF | 2.111  | 2.483  | 0.000  | 0.000  |
> | Visual_DIFF | 3.474  | 1.121  | 0.000  | 0.000  |
> | OF_DIFF | 1.998  | 0.003  | 1.476  | 0.239  |
>
> The pairwise analysis revealed that vision is the strongest modality, as its unique interactions consistently surpass those of other modalities. Furthermore, we observed a primarily redundant relationship between vision and frame difference, while frame difference and optical flow demonstrated notable synergy. This observed synergy likely arises because frame difference and optical flow, when considered individually, carry less task-relevant information and thus benefit significantly from combining their complementary information to achieve better task performance.
>
>
> **Q3. physical meanings of $i(x_1;x_2; y)<0$.**
>
> A3: Thank you for raising this important question. To build intuition, consider the simpler case of negative unimodal information:
> $i(x_1;y) = \log\frac{p(y|x_1)}{p(y)}$.
> Negativity implies $p(y|x_1) < p(y)$, meaning that observing modality $x_1$ decreases the likelihood of the target $y$ compared to its prior probability. This suggests $x_1$ provides information that contradicts $y$, a situation potentially arising from mislabeled samples.
>
> For the interaction term, defined as:
> $i(x_1;x_2;y) = i(x_1;y) + i(x_2;y) - i(x_1,x_2;y)$
> a negative value ($i(x_1;x_2;y) < 0$) signifies that the joint information $i(x_1,x_2;y)$ is greater than the sum of the individual information components ($i(x_1;y) + i(x_2;y)$). This phenomenon represents synergy: the modalities $x_1$ and $x_2$, when combined, provide more information about $y$ (or potentially, more "mis-information" if the individual terms are strongly negative) than would be expected from merely summing their individual contributions.
>
> Regarding the redundancy term $r(x_1;x_2;y)$ discussed in our paper (which forms part of the unimodal information, e.g., $i(x_1;y) = r + u_1$), this can also be negative. This is particularly likely if both individual modalities carry contradictory information about the label (i.e., $i(x_1;y) < 0$ and $i(x_2;y) < 0$). In such cases, the redundant information component may also reflect this contradictory concerning the target $y$.
>
> [1] Tobias Mages et al. "Decomposing and Tracing Mutual Information by Quantifying Reachable Decision Regions." Entropy, 2024.
>
> [2] Paul Williams et al. "Nonnegative Decomposition of Multivariate Information." arXiv, 2010.
>
> [3] Paul Pu Liang et al. "Quantifying & Modeling Multimodal Interactions: An Information Decomposition Framework." NeurIPS, 2023.

---

> > ### Comment · Reviewer_sN2h · 2025-04-04
> >
> > For Q1, the author seems to give the visualization results of the entire dataset. However, the reviewer is curious about the visualization results on a single sample, which is the core contribution of the paper. For example, taking an audio-vision sample on the CRAME-D dataset as an example, how do its r, s, and u change when adding noise to the vision modality
> >
> > For Q2, why not use CMU-MOSEI for verification? It naturally includes three modalities: audio, vision, and text. The frame difference and optical flow seem to be the same modality (the same physical meaning)?

---

> > > ### Author Response · Authors · 2025-04-06
> > >
> > > We thank the reviewer for their valuable suggestions and positive feedback. Here are our responses:
> > >
> > > **Q1: Visualization of sample-level interaction with noise.**
> > >
> > > A1: Thank you for this valuable suggestion. We have analyzed the impact of noise on interactions at the sample level. Below are results for representative samples, comparing clean conditions to conditions where noise was added to one modality (assumed visual, affecting $\tilde{u}_v$):
> > >
> > > |  | Clean |  |  |  | Noisy |  |  |  |
> > > |---|:---:|:---:|:---:|:---:|:---:|:---:|:---:|:---:|
> > > | Sample | $r$ | $u_v$ | $u_a$ | $s$ | $r$ | $\tilde{u}_v$ | $u_a$ | $s$ |
> > > | 1 | 0.652  | 0.000  | -0.309  | 0.723  | 0.343  | -0.127  | 0.000  | 0.500  |
> > > | 2 | 0.064  | 0.000  | -0.255  | 0.900  | -0.191  | -0.385  | 0.000  | 1.316  |
> > > | 3 | 0.965  | 0.000  | 0.805  | 0.008  | 1.769  | -1.453  | 0.000  | 1.470  |
> > > | 4 | 0.424  | 0.000  | -1.215  | 2.299  | -0.790  | -0.736  | 0.000  | 2.753  |
> > >
> > > From the table above, we can observe that when noise is applied to a single modality, its unimodal mutual information ($u_v + r$) decreases. Additionally, since noise increases the entropy within the modality, introducing greater uncertainty to the visual modality, the calculated values of $r^+$ and $r^-$ also change. This leads to $u_a$ becoming 0 after noise addition, while $\tilde{u}_v$ becomes negative. Our method compares samples that are relatively robust to noise sensitivity (samples 1 and 2) with those that are more sensitive (samples 3 and 4), clearly highlighting the changes in their interactions.
> > >
> > > We will add this analysis of sample-level interaction with noise in the revised manuscript.
> > >
> > > **Q2: CMU-MOSEI for verification.**
> > >
> > > A2: Thank you for your valuable question. As you mentioned, the UCF dataset is characterized by certain correlations among its three modalities (video, optical flow, and frame difference): video provides rich visual information, while frame difference and optical flow are derived from the video modality, modeling and describing the dynamic changes in the video. This experiment effectively demonstrates the interaction effects when different modalities are correlated.
> > >
> > > We have also conducted experiments on the CMU-MOSEI dataset (Vision+Audio+Text), which presents a case where the three modalities have weaker correlations but jointly describe the same entity, as shown in the Table below.
> > >
> > > | modalityPair | $R$ | $U_1$ | $U_2$ | $S$ |
> > > |:---:|:---:|:---:|:---:|:---:|
> > > | V+T | 0.121  | 0.000  | 0.163  | 0.005  |
> > > | V+A | 0.116  | 0.010  | 0.000  | 0.012  |
> > > | A+T | 0.127  | 0.000  | 0.248  | 0.002  |
> > >
> > > Our experimental results reveal that among these three modalities, Text serves as the primary modality with strong unique characteristics, while Vision and Audio modalities exhibit lower discriminative power individually but contribute more synergistic information when combined. We will include these three-modality experiments and a comprehensive discussion of the results in our revised manuscript.
> > >
> > > We sincerely thank the reviewers for their thorough review and thoughtful responses. Your suggestions have significantly enhanced the quality of our paper, and we will revise and refine our manuscript according to your valuable feedback.

---

### Official Review · Reviewer_G7u5 · 2025-03-16

**Overall Recommendation:** 2

**Summary:**

The paper tackles an important question of capturing interactions between modalities and proposes a lightweight
entropy-based multimodal interaction estimation approach for efficient and precise sample-wise interaction measurement across various continuous distributions. The authors  demonstrate the efficacy of their approach using circuit logic  recovery and experimentation with real datasets.

**Claims And Evidence:**

The authors claim that the definition of redundancy at the event (sample) level is clearer and more straightforward than that of uniqueness or synergy. Hence they suggest to obtain redundant information by partitioning mutual information into components relative to the target, allowing for the measurement suitable for redundancy in each component. They name their method Lightweight Sample wise Multimodal Interaction (LSMI).

**Essential References Not Discussed:**

I'm only familiar with L.P. Morency works on capturing  multimodal interactions, and those were cited.

**Experimental Designs Or Analyses:**

I feel that paper will greatly benefit from a more precise discussion are its contribution. It might help to identify specific cases  where their "sample" level redundancy estimation works better and motivate the readers with those cases before providing the numbers. Also it will help to state in Table 1 if lower or higher numbers are better.

**Methods And Evaluation Criteria:**

The authors evaluate on synthetic and real world datasets (KS dataset, UCF dataset, CREMA-D dataset, KS dataset) and compare to several other methods (Weighted ensemble vs. LSMI ensemble  and PID method). However the conclusions are relatively not impresive:
"The comparison results are shown in Table 2. We observe that our LSMI estimator is largely consistent with the PID-Batch
method. Furthermore, in the KS, Food-101, and MOSEI datasets, our method aligns with the top two highest interactions in terms of redundancy and uniqueness, respectively."

**Other Comments Or Suggestions:**

Please rewrite the paper, clearly stating the contribution and motivate the sample-level estimation of interactions. It will be helpful to see how the findings can be used in downstream tasks like ImageBind or even MLLM training.

**Other Strengths And Weaknesses:**

In overall, it was really hard to follow the  paper and its specific contributions. A year ago I was very impressed by the work from Morency's lab on  capturing the multimodal interactions, but this paper does not contribute enough on top of the original work (PID algorithm they compare to).

**Questions For Authors:**

None

**Relation To Broader Scientific Literature:**

I think the topic of capturing multimodal interactions is extremely important for a broader community that aims to develop MLLMs and align modalities. Understanding interaction between training samples' modalities might improve the training purpose. Unfortunately this paper does not  go beyond estimation of interactions.

**Theoretical Claims:**

I do not think the paper has any theoretical claims

---

> ### Author Rebuttal · Authors · 2025-04-01
>
> We thank the reviewers for their valuable suggestions. Here are our responses to the questions raised about our paper.
>
> **Q1: The contribution of this work compared to the work from Morency's Lab.**
>
> A1: Thank you for this question. The core contribution of our work, compared to foundational studies from Morency's Lab [1], is the development of efficient sample-wise quantification of multimodal interactions.
>
> We build upon the same established interaction concepts (redundancy, uniqueness, synergy) common to both [1] and other literature [2,3]. However, the key distinction lies in mathematical modeling of interactions:
> Morency's work (PID) [1] typically provides dataset-level aggregate measures of interactions.
> Our method enables quantifying these interactions for **each individual sample**.
>
> This efficient sample-level resolution is our main technical contribution and is vital for applications requiring instance-specific insights, such as targeted model integration or knowledge distillation strategies explored in our experiments (Section 4.3). Aggregate measures (PID) cannot readily support these tasks.
>
> Notably, the value of such pointwise measures was recognized as an important next step by Paul and Morency 2023: "*A natural extension is to design pointwise measures... which could help for more fine-grained dataset and model quantification...*" Our work provides a concrete method to realize this, demonstrating its contribution and relevance.
>
> **Q2: Application of LSMI to Downstream Tasks**
>
> Thank you for the suggestion. Our LSMI method enables the estimation of sample-level interactions, which can indeed benefit downstream tasks beyond the model ensemble and distillation examples discussed in the main paper.
>
> To demonstrate this, we applied LSMI to guide the fine-tuning process for a specific task. We used the ImageBind model, which focuses on modality alignment, and fine-tuned it on the Kinetics-Sounds (KS) dataset (Audio+Visual). Specifically, we employed LSMI to quantify the degree of redundancy between the audio and visual modalities for each sample. Based on this metric, we partitioned the KS dataset into two subsets: one containing samples with relatively high redundancy ($R$) and another with relatively low redundancy ($U+S$, comprising samples dominated by unique or synergistic information).
>
> We then fine-tuned the ImageBind model separately on these two subsets ($R$ and $U+S$) and compared the results against fine-tuning on the entire dataset ('alldata'). The performance, evaluated by classification accuracy on KS dataset, is shown below:
>
> | Fine-tuning Set | V+A Performance | V Performance | A Performance |
> |-----------------|-----------------|---------------|---------------|
> | alldata         | 0.8539          | 0.8183        | 0.7270        |
> | U+S             | 0.8496          | 0.8045        | 0.7289        |
> | R               | 0.8772          | 0.8241        | 0.7257        |
>
> As observed, fine-tuning specifically on the high-redundancy subset ($R$) identified by LSMI yielded the best performance for the combined modalities (V+A) and the visual modality (V). This suggests that leveraging LSMI to identify and focus on highly redundant samples can be an effective strategy for enhancing model fine-tuning, potentially by reinforcing learning when sufficient information is present in individual modalities.
>
>
> **Q3:  Result state in Table 1**
>
> A3: We appreciate you pointing it out. Table 1 focuses on evaluating the precision of different interaction estimation methods. We utilize synthetic datasets based on logic relations with additive noise, which allows for comparison against known ground truth (GT) interaction values. This discrete logic carries explicit interaction information and has been widely used in previous studies [1,4]. The objective is to assess **how closely each estimator's output matches the GT**. As the results indicate, our method achieves interaction estimates that align more closely with the GT compared to the baseline estimators evaluated. This finding underscores the robustness of our approach, demonstrating its effectiveness in capturing underlying data interactions despite the presence of noise. We have enhanced the description surrounding Table 1 in the revised manuscript to better emphasize this point.
>
> [1] Paul Pu Liang et al. "Quantifying & Modeling Multimodal Interactions: An Information Decomposition Framework." NeurIPS, 2023.
>
> [2] Benoit Dufumier et al. "What to align in multimodal contrastive learning?" ICLR, 2025.
>
> [3] Paul Pu Liang et al. "Multimodal learning without labeled multimodal data: Guarantees and applications." ICLR, 2024.
>
> [4] Nils Bertschinger et al. "Quantifying Unique Information." Entropy, 2014.

---

> > ### Comment · Reviewer_G7u5 · 2025-04-05
> >
> > I thank the authors for the detailed responses to my comments and other comments. In overall the contribution is more clear to me (a direct extension, following Paul and Morency's suggestion). However,  I think that a real contribution should come in the application to real world problems.  I would assume that the benefit should be from synergy between the modalities, and I'm confused regarding the above mentioned ImageBind experiment, similar to wohR's
> > Q6: Appreciate the additional experiment. Could you explain the results here? As I understand, the overall performance increases when redundant data is employed, rather than unique and synergistic samples. This seems counter-intuitive.
> >
> > Therefore, while I slightly raised my rating, I cannot suggest to accept this paper.

---

> > > ### Author Response · Authors · 2025-04-06
> > >
> > > Thank you for your valuable comment. Please find our detailed response below.
> > >
> > > **Q: Explanation of ImageBind experiment.**
> > >
> > > A: Thank you for your valuable question regarding the ImageBind experiment results. The key reason lies in the contrastive learning mechanism used for fine-tuning ImageBind in our experiments. This method aims to align representations by pulling positive pairs (inputs assumed to represent the same concept) closer together in the embedding space. For this alignment to be effective, the fundamental assumption is that the paired modalities contain consistent or shared information relevant to the alignment goal.
> > >
> > > In our setup, we specifically fine-tuned the pre-trained ImageBind model using contrastive learning on subsets of the KS dataset, targeting improved modal alignment for that specific downstream task. We then evaluated the effectiveness of this alignment (and the quality of the fine-tuned features) by training a simple linear layer on the features to perform the task. Within this context, we can analyze data based on modal interaction:
> > >
> > > Data with predominantly Redundant interaction: These pairs naturally fulfill the consistency requirement for contrastive learning. Both modalities convey similar or overlapping information relevant to the task. Using these pairs for contrastive fine-tuning effectively strengthens the alignment based on shared semantics, leading to better features for the downstream linear evaluation.
> > >
> > > Data with predominantly Unique or Synergistic interaction: By definition, these pairs contain information that is significantly different, inconsistent, or emerges only through combination across the modalities. Attempting to force alignment between these inherently less consistent representations via a contrastive loss introduces noise or conflicting optimization signals. This hinders the specific fine-tuning process focused on strengthening existing semantic alignment, potentially degrading performance compared to using more consistent data or the original pre-trained model.
> > >
> > > Our LSMI method serves precisely to identify data pairs exhibiting high task-specific redundancy (information consistency). The experiment demonstrates that selecting such data improves the contrastive fine-tuning outcome for ImageBind because it provides the consistent positive pairs this particular learning method leverages most effectively.
> > >
> > > Therefore, the result highlights a specific property of the contrastive alignment technique itself during fine-tuning: it benefits more from informational consistency (redundancy) in the training pairs.
> > >
> > > Regarding synergistic interaction, it typically describes cases where task-relevant information emerges from combining modalities, even if the individual modalities are insufficient on their own. Contrastive alignment methods, like those in ImageBind pre-training/fine-tuning, are primarily designed to map existing semantic similarities into a shared space. They are inherently less suited to capturing this emergent information that arises purely from the combination in synergistic cases, as forcing alignment between representations without strong pre-existing semantic overlap can be counterproductive. Our experiments, showing degraded performance when fine-tuning on less consistent data, support this distinction.
> > >
> > >
> > > Thank you again for your time and valuable feedback.

---

### Official Review · Reviewer_wohR · 2025-03-16

**Overall Recommendation:** 2

**Summary:**

The paper presents LMSI, a light-weight framework for estimating redundant, unique and synergistic information in multimodal tasks through sample-level partial information decomposition. While the previous approaches require computationally expensive tasks like distribution estimation, LMSI is cheaper without compromising on the accuracy.  Further, LMSI is positive, monotonic and is grounded information theory making it attractive to use. The authors supplement LMSI with many experiments, and against real-world application of model distillation.

#####update after the rebuttal######

I thank the authors for engaging in the rebuttal and trying to answer my queries. However, I still have my reservations about the ImageBind and OOD experiments. The authors make some very bold and interesting claims about the role of synergistic and redundant interactions in the paper. These claims might be even correct but currently stand unsubstantiated.  Since the work leaves some critical ends loose, I maintain my initial score.

##############################

**Claims And Evidence:**

Multimodal biases arise from (1) dataset only, (2) model only, and (3) both (referred to as categories below). \
First the authors test the validity of their method on simulated dataset (section 4.1) which falls in the 2nd category. Next, in section 4.2.2 (Dataset Interactions), the authors quote in L354-358 "we select models with the lowest generalization risk, which are considered to better model the underlying distribution, and average their sample-level interactions to represent the dataset-level interactions."  I am not convinced by this. While the dataset might have biases, the model might not be necessarily relying on those biases (that is, on the same dataset, we can have both biased and unbiased models [1]). Therefore, section 4.2.2 is actually a dataset+model (category 3). However, the section is misleadingly titled as "Dataset Interactions", and the discussion on model interactions is absent in the paper. Perhaps a better method to achieve sample-level interactions would be to use LMSI on (1) an overfit model or (2) ensemble of models of different families, to ameliorate the effect of model's biases.


References: \
[1] Rawal, Ishaan Singh, et al. "Dissecting multimodality in VideoQA transformer models by impairing modality fusion." ICML 2024

**Essential References Not Discussed:**

1. QUAG [1] which was also used for "multimodal interaction quantification at the sample level for real-world data"
2. Comparison against and discussion of PID [2]

References: \
[1] Rawal, Ishaan Singh, et al. "Dissecting multimodality in VideoQA transformer models by impairing modality fusion." ICML 2024 \
[2] Liang, Paul Pu, et al. "Quantifying & modeling multimodal interactions: An information decomposition framework." NeurIPS 2023

**Experimental Designs Or Analyses:**

The methods section of the paper lacks many details. The implementation details, model details, details of the finetuning experiment are missing.

**Methods And Evaluation Criteria:**

Can this method be used for more concrete tasks to validate its correctness? For example, does LMSI corroborate with the findings like in [1] and [2] where the authors explicitly propose debiased datasets and models (that is, decrease the redundant information in the model and dataset?)


References: \
[1] Buch, Shyamal, et al. "Revisiting the" video" in video-language understanding." CVPR 2022. \
[2] Goyal, Yash, et al. "Making the v in vqa matter: Elevating the role of image understanding in visual question answering." CVPR 2017.

**Other Comments Or Suggestions:**

The supplementary section can be improved with more examples and details.

**Other Strengths And Weaknesses:**

**Strengths:**
1. The paper tackles an important task of multimodal interaction quantification
2. LMSI is light-weight and accurate
3. LMSI is grounded in theoretical and well-founded ideas in information theory


**Weaknesses:**
1. Overarching claims about dataset biases without considering model biases (see Claims And Evidence)
2. Lack of confirmatory real-world experiments (see Methods And Evaluation Criteria)
3. No comparison against attention-based methods (early, mid and late fusion strategies)

I believe that the work is headed in the right direction however it could be misinterpreted because of lack of discussions and supporting experiments.

**Questions For Authors:**

1. What is the correlation between human judgement and LMSI-predicted interations in Table 2?
2. How does LMSI account for noise/missing information? Labelling process is not perfect, especially for subjective or (semi)-automatic labelling tasks?
3. How to make sense of absolute LMSI score? Can it be consolidated into a single number? Ideally, a measure that has well-behaved bounds. Because currently, I am not sure if u=0.1 is good or bad?

**Relation To Broader Scientific Literature:**

The work is quite important and can have a lot of different applications in dataset collection, filtering and interpretability tasks.

**Theoretical Claims:**

Yes, LMSI is grounded in fundamental ideas of information theory

---

> ### Author Rebuttal · Authors · 2025-04-01
>
> Thank you for your valuable comments and suggestions. Here are our responses:
>
> **Q1: Model biases in dataset interactions**
>
> A1: We appreciate the reviewer's valuable point. Indeed, quantifying multimodal interactions requires models, making the estimates inherently model-dependent. Consistent with prior work yielding reliable outcomes [1], our method employs fully trained multimodal models. We focused on models demonstrating strong generalization, as we posit these better reflect the underlying data distribution and better modelling the interactions within it.
>
> Following the reviewer's suggestion, we constructed an ensemble integrating several of best-generalizing models to measure interactions. As detailed in [Table a](https://anonymous.4open.science/r/Submission2900-7FB8/Tab_a.png), the interaction estimates derived from this ensemble were highly consistent with those from individual top-performing models, exhibiting negligible differences.
>
> **Q2: Comparison against attention-based methods**
>
> A2: Thank you for this valuable suggestion. We validated LMSI on a Multimodal Transformer with hierarchical attention (from multi-stream to single-stream)[2], where we varied the layer $l$ at which cross-modal fusion was first introduced. A smaller $l$ indicates earlier fusion architecture.
> The experiment results are shown in [Table b](https://anonymous.4open.science/r/Submission2900-7FB8/Tab_b.png).
> Our analysis revealed that interactions learned from later fusion stages were predominantly characterized by redundancy, whereas interactions learned from earlier fusion stages demonstrated a stronger tendency to capture synergistic relationships between modalities.
>
> **Q3: Understand the absolute scores of LMSI**
>
> A3: Thank you for this insightful question. LMSI components ($r$, $s$, $u_1$, $u_2$) measure information quantity (in nats/bits) from different multimodal interactions. We can utilize relative proportions (e.g., $\hat{r} = \frac{r}{i(x_1,x_2;y)}$) as a bounded, standardized metric. These relative scores ($\hat{r}$, $\hat{s}$, $\hat{u}_1$, $\hat{u}_2$) sum to 1 and provide a normalized interaction view. The optimal values of interaction varies by task - video tasks show high $\hat{r}$, while sentiment analysis favors $\hat{u}$, as shown in Table 2 in the main paper.
>
>
> **Q4: Comparison with human judgment**
>
> A4: Thank you for this point. Direct numerical comparison is challenging because human evaluations are typically sample-specific and inherently subjective, making it difficult to accurately represent the information quantity across the entire data distribution. Despite these differences, human judgments can provide valuable insights into the perceived relative importance or ranking of interactions within specific instances. Following precedents like [1], we investigated the alignment between LMSI and human perception by focusing on the dominant interaction types identified by both. As shown in Table 2, LMSI demonstrates reasonable consistency with human assessments in identifying top interactions across datasets.
>
>
> **Q5: LSMI Behavior with Noise**
>
> A5: Thank you for your question. LSMI quantifies task information $Y$ across modalities $(x_1, x_2)$, making it sensitive to information quality. Label noise reduces the multimodal mutual information $i(x_1, x_2; y)$, affecting LSMI's interaction measures. We added noise to image-text pairs [Table c](https://anonymous.4open.science/r/Submission2900-7FB8/Tab_c.png), observing decreased interactions, especially for information-rich samples. This confirms LSMI's sensitivity to noise.
>
> **Q6: Request for concrete tasks**
>
> A6: Thank you for this suggestion. We wish to clarify that our method, LMSI, has already been validated on several concrete tasks: sentiment recognition (MOSEI), action recognition (KS), and sarcasm detection (URfunny).
>
> To further demonstrate LSMI's practical utility, we applied it to guide ImageBind fine-tuning on the KS dataset (Audio+Visual modality). Using LSMI, we quantified sample-wise audio-visual redundancy and partitioned the KS dataset into two subsets: high-redundancy ($R$) and low-redundancy ($U+S$). We then fine-tuned ImageBind separately on these subsets as well as the full dataset ('alldata'), comparing their classification performance. The results, presented in [Table d](https://anonymous.4open.science/r/Submission2900-7FB8/Tab_d.png), demonstrate that our data partitioning strategy based on LSMI analysis can enhance fine-tuning outcomes. While our VQA task experiments are still in progress due to time limitations, these results highlight the potential of LSMI for improving model fine-tuning.
>
> [1] Paul Pu Liang et al. "Quantifying & Modeling Multimodal Interactions: An Information Decomposition Framework." NeurIPS, 2023.
>
> [2] Peng Xu et al. "Multimodal Learning with Transformers: A Survey." TPAMI, 2023.

---

> > ### Comment · Reviewer_wohR · 2025-04-04
> >
> > Thank you for the clarifications. I went through the rebuttal and the additional experiments. Some follow-up comments:
> >
> > Q2: I thank the authors for adding experiments for early, mid and late fusion. The finding should add value to the paper.
> >
> > Q3: I doubt if the relative proportions would be really informative in comparing the models. The individual values still might be very "off" across different models, and I am not sure if this is a good way of comparing the models.
> >
> > Q4: Since human evaluations and the proposed method both are based on sample-level statistics, I am unable to understand why is correlation across across the evaluations has not been reported?
> >
> > Q5: Thanks for adding these results. Could you clarify why did you select samples with u_2 = 0? Perhaps these findings are confounded in unimodal biases of the model?
> >
> > Q6: Appreciate the additional experiment. Could you explain the results here? As I understand, the overall performance increases when redundant data is employed, rather than unique and synergistic samples. This seems counter-intuitive.
> >
> > I still believe that the work has its own merits but could benefit from more refinement with  zero-shot OOD evaluations.

---

> > > ### Author Response · Authors · 2025-04-07
> > >
> > > We truly appreciate your recognition of our work, and we think that the questions you posed can indeed help to polish our core contributions a lot. Here are our point-to-point responses, hoping them can well address your concerns:
> > >
> > > **Q3: Metric for comparing models.**
> > >
> > > Thank you for clarifying. Our method supports fine-grained model comparison at the sample level by estimating information contributions across interaction types. To compare models A and B, we: (1) compute interaction vectors $V^A = (r^A, u_1^A, u_2^A, s^A)$ and $V^B = (r^B, u_1^B, u_2^B, s^B)$ for each sample, (2) calculate the distance between these vectors, and (3) average distances across samples.
> > >
> > > These sample-level differences are utilized in our interaction-based ensemble method (Section 4.3.2), where smaller distances receive higher fusion weights, improving ensemble performance. Results in [Figure b](https://anonymous.4open.science/r/Submission2900-7FB8/Fig_b.png) show interaction similarities between models across CREMAD datasets.
> > >
> > > While this provides sample-level analysis, the 'relative proportions' metric discussed previously offers dataset-level summaries of model preferences for interaction types. Our framework thus enables evaluation at multiple granularities: detailed sample-level comparisons between models and characterization of overall model biases in multimodal information utilization.
> > >
> > > **Q4: Correlation with human study.**
> > >
> > > Thank you for this suggestion. Following the suggestion, we analyzed the correlation between sample-level LSMI scores and corresponding human evaluations on the Food-101 dataset. This dataset is selected as a baseline that achieves high performance on it, and its interaction modeling is relatively accurate. We found strong Pearson correlation coefficients between LSMI and human scores: 0.98 for redundancy and 0.95 for text uniqueness. These results indicate that LSMI aligns well with human judgments regarding interaction qualities.
> > >
> > > **Q5: Reason for $u_2 = 0$.**
> > >
> > > Thank you for your question. In our Food-101 experiment, the text modality typically contains more information than images and has higher entropy, resulting in most samples having $u_2=0$. We also analyzed samples with $u_2>0$, with results in [Table c](https://anonymous.4open.science/r/Submission2900-7FB8/Tab_c.png) showing how unique image information responds to label noise.
> > >
> > > **Q6: Experimental result about ImageBind.**
> > >
> > > Thank you for your valuable question. We fine-tune ImageBind using a contrastive loss specifically to demonstrate how LSMI can enhance targeted fine-tuning through strategic data selection. Contrastive learning performs best when aligning modalities with shared information (i.e., 'redundant' data), which improves representation quality and downstream performance. In contrast, with 'unique' or 'synergistic' data, modalities lack sufficient consistency. Forcing alignment between these inconsistent pairs with minimal semantic overlap introduces optimization noise that can degrade rather than enhance representation quality.
> > >
> > > LSMI serves as an effective data selection mechanism by precisely identifying samples with high inter-modal consistency (redundancy). By fine-tuning ImageBind using contrastive loss exclusively on the data subset selected by LSMI, we concentrate the beneficial contrastive alignment process on the most suitable samples. Our experimental results convincingly demonstrate the effectiveness of this targeted approach, showing significantly improved performance when fine-tuning on the LSMI-selected subset compared to scenarios involving less consistent data or random selection strategies.
> > >
> > > **Q7: OOD evaluation.**
> > >
> > > Thank you for suggesting OOD scenarios as a valuable validation approach for our estimator. Following [1], we conducted OOD evaluation using the Multimodal Near-OOD protocol on UCF and Kinetics-Sounds (KS) datasets, training on in-domain (ID) data and evaluating interactions on both ID and OOD data. As shown in [Table e](https://anonymous.4open.science/r/Submission2900-7FB8/Tab_e.png), under OOD conditions, total information decreases, redundancy significantly reduces, while synergy increases. This implies that in OOD scenarios, shared information between modalities becomes less reliable while distributions shift from training data, reducing redundancy. Simultaneously, increased synergy indicates that the model leverages more complementary interactions. This shift from redundancy-dominated to synergy-dominated processing represents a key adaptation mechanism for generalization. We will include these discussion in the revised version.
> > >
> > > We sincerely appreciate your valuable suggestions and insightful feedback. Your suggested refinements indeed helped us better polish the value of our paper. We are truly grateful for your assistance!
> > >
> > > [1] Dong, H. "MultiOOD: Scaling Out-of-Distribution Detection for Multiple Modalities." NeurIPS 2024.

---

### Decision · Program_Chairs · 2025-05-01

**Decision:**

Accept (poster)

**Comment:**

__Summary__

This submission presents a framework for capturing (and estimating) three types of interactions between two data modalities for a learning task: redundancy, uniqueness, and synergy, all under an information-theoretic framework. Previous work on multi-modal interactions relies on density estimation, which is both computationally demanding and becomes more and more challenging and inaccurate as dimensionality increases. The novelty in the framework is twofold:

1. It estimates the three types of interactions at the sample level through partial information decomposition, and as a byproduct, the framework is computationally efficient as opposed to previous methods.

2. The framework resolves conflicts between pointwise mutual information (PMI) and partial information decomposition (PID) axioms in multi-modal learning (non-negativity, monotonicity), which are usually defined at the distribution level.

The authors demonstrate the efficacy of the proposed framework using circuit logic recovery (synthetic data) and various experiments with real datasets. In addition, they presented additional experimental results based on reviewers' requests.

All reviewers agree that the paper proposes an interesting and important step for estimating multi-modal interactions, which can, in turn, have different applications in multi-modal ensemble learning, distillation, dataset collection, filtering, and interpretability.

__Regarding reviewers' concerns, authors' rebuttal, and reviewers' final response to authors' rebuttal.__

For Rev1 (wohR): The authors have addressed all concerns raised by the reviewer in two rounds of questions. Unfortunately, the reviewer did not respond to the last reply from the authors, although I asked the reviewer to provide feedback to the authors. The last concerns/questions raised by the reviewer are secondary or do not really affect the paper's core ideas and contributions.

For Rev2 (G7u5): The authors have addressed all concerns raised by the reviewer in two rounds. The specific concern raised by the reviewer regarding the ImageBind experiment was also addressed in detail in the second author's rebuttal. The authors also explained how the nature of contrastive learning takes advantage of the redundancy in training pairs, which explains the results in the Table provided by the authors. The reviewer did not give feedback to the authors.

Rev3 (sN2h) and Rev4 (r5Mr) have no issues with the submission, and the authors addressed the questions raised by Rev3.